# The Four-Parameter PSS Method for Solving the Sylvester Equation

**Hai-Long Shen, Yan-Ran Li * and Xin-Hui Shao**

Department of Mathematics, College of Sciences, Northeastern University, Shenyang 110819, China; hailong_shen@126.com (H.-L.S.); shaoxinhui@mail.neu.edu.cn (X.-H.S.)
* Correspondence: 1800116@stu.neu.edu.cn

**Abstract:** In order to solve the Sylvester equations more efficiently, a new four parameters positive and skew-Hermitian splitting (FPPSS) iterative method is proposed in this paper based on the previous research of the positive and skew-Hermitian splitting (PSS) iterative method. We prove that when coefficient matrix $A$ and $B$ satisfy certain conditions, the FPPSS iterative method is convergent in the parameter's value region. The numerical experiment results show that compared with previous iterative method, the FPPSS iterative method is more effective in terms of iteration number IT and runtime.

**Keywords:** Sylvester equation; Positive and skew-Hermitian iterative method; FPPSS iterative method

## 1. Introduction

In this paper, we mainly consider the problem of solving the continuous Sylvester equations with the following form:

$$AX + XB = C, \tag{1}$$

where $A \in \mathbb{C}^{m \times m}$, $B \in \mathbb{C}^{n \times n}$, $C \in \mathbb{C}^{m \times n}$ are given matrices that satisfy the following conditions:

(I)   $A$, $B$, and $C$ are large-scale and sparse matrices;
(II)   At least one of $A$ and $B$ is a non-Hermitian matrix;
(III)   At least one of the positive semidefinite matrices $A$ and $B$ is a positive definite matrix.

The solution of Equation (1) exists and is unique.

This kind of matrix equation has a wide range of applications in scientific computing and engineering fields. Problems like digital image restoration, control systems, electromagnetic field processing, neural networks, and model reduction will eventually involve the solution of large-scale Sylvester equations [1–3]. Because the time required to solve the Sylvester equation is related to the speed of solving actual problems, designing an effective method for solving the Sylvester equation is a subject with theoretical research and practical application value.

In the past few decades, scholars have focused on the methods of solving such problems. Therefore, more and more direct and iterative solutions are proposed. However, because the coefficient matrix of the equation to be solved is mostly a large and sparse matrix, the direct method is not applicable compared with the iterative method. In 1952, the conjugate gradient method (CG) was proposed to solve symmetric positive definite linear equations [4].

In 1986, in order to solve the problem of asymmetric coefficient matrix, Saad. Y et al. put forward the famous Generalized Minimal Residual (GMRES) algorithm which has better stability and less storage space than the previous Krylov subspace algorithm [5].

In 2003, Bai Zhongzhi et al. proposed the Hermitian and skew-Hermitian splitting iterative method, namely HSS iterative method [6]. After that, many academicians at home and abroad have improved this kind of method, such as the method based on positive definite and skew-Hermitian splitting of coefficient matrix, i.e., the PSS iteration method [7]; the NSS iteration method in views of normal and skew-Hermitian splitting [8]; and according to various preconditioning technique, the preconditioned HSS iterative method [9,10], lopsided HSS iterative method [11], modified generalization HSS iterative method [12], and so on.

The HSS iterative method and its variants have many mature and effective extensions to solve continuous Sylvester equation.

In 2011, based on the Hermitian splitting and skew-Hermitian splitting of coefficient matrices, Bai et al. applied HSS iteration method to solve continuous Sylvester equation for the first time [6].

In 2013, Wang Xiang and others solved Sylvester equation by PSS iteration method [13].

In 2014, Zheng Qingqing and others used NSS iteration method to solve Sylvester equation [14].

In 2015, MHSS iteration method and GHSS iteration method were proposed successively [15,16].

In 2017, PMHSS iteration method was proposed [17].

It can be seen that most of the methods for solving Sylvester equation are improved and generalized based on HSS iteration method and there is still room for research on the promotion and application of PSS algorithms. Based on the above reasons, in order to further improve the solving speed of Sylvester equation, a new four-parameter PSS iteration method, namely FPPSS iteration method is proposed to solve the continuous Sylvester equation. The parameters that minimize the upper bound of the spectral radius of the iteration matrix are derived, and the effectiveness and stability of the iteration method are proved by numerical experiments.

The structure in this paper is as follows. In Section 2, the iterative scheme of the FPPSS iterative method for solving the large-scale continuous Sylvester equation with non-Hermitian positive definite/semidefinite matrix is given, and the exact range of parameters for guaranteeing the convergence of the FPPSS iterative method is theoretically calculated. Moreover, optimal iterative parameters that bring the upper bound of the spectral radius of the iterative matrix to a minimum are derived. In Section 3, numerical experiments compare the FPPSS iterative method with the PSS iterative method to demonstrate the effectiveness and stability of FPPSS. Finally, in Section 4, some conclusions are given.

## 2. The Four-Parameter PSS Iterative Method

In order to further improve the convergence speed of the PSS iterative method, a four-parameter PSS iterative method, namely FPPSS iterative method, is proposed to solve the continuous Sylvester equation.

Now, we use $P(V)$ and $S(V)$ to represent the positive and skew-Hermitian part of matrix $V \in \mathbb{C}^{n \times n}$, respectively. Obviously, matrix $V$ has positive definite and skew-Hermitian splitting, i.e., PSS iterative method [7]:

$$V = P(V) + S(V).$$

Analogy to the PSS method, the matrix $A$ and $B$ have the following forms of splitting:

$$A = (\alpha_1 I + P(A)) - (\alpha_1 I - S(A)) = (\beta_1 I + S(A)) - (\beta_1 I - P(A)),$$
$$B = (\alpha_2 I + P(B)) - (\alpha_2 I - S(B)) = (\beta_2 I + S(B)) - (\beta_2 I - P(B)),$$

where $\alpha_j (j = 1, 2)$ are given non-negative constants and $\beta_j (j = 1, 2)$ are positive constants, $I$ is the identity matrix with the appropriate dimension.

Then Equation (1) can be equivalently rewritten as:

$$\begin{cases} (\alpha_1 I + P(A))X + X(\alpha_2 I + P(B)) = (\alpha_1 I - S(A))X + X(\alpha_2 I - S(B)) + C, \\ (\beta_1 I + S(A))X + X(\beta_2 I + S(B)) = (\beta_1 I - P(A))X + X(\beta_2 I - P(B)) + C. \end{cases}$$

In the assumption (I)–(III), we can observe that matrices $\alpha_1 I + P(A)$ and $-(\alpha_2 I + P(B))$ have no common eigenvalues, while matrices $\beta_1 I + S(A)$ and $-(\beta_2 I + S(B))$ also have no common eigenvalues, so the above two equations have a unique solution for any given right end, which results in the following four-parameter positive definite and skew-Hermitian splitting iterative method for solving the continuous Sylvester Equation (1), namely the FPPSS iterative method.

**Theorem 1.** *Given any initial matrix $X^{(0)} \in \mathbb{C}^{m \times n}$, for $k = 0, 1, 2, \ldots$, $X^{(k+1)} \in \mathbb{C}^{m \times n}$ is calculated in the following format until the iteration sequence $\left\{ X^{(k)} \right\}_{k=0}^{\infty}$ satisfies the convergence condition:*

$$
\begin{cases}
(\alpha_1 I + P(A))X^{(k+\frac{1}{2})} + X^{(k+\frac{1}{2})}(\alpha_2 I + P(B)) \\
\qquad = (\alpha_1 I - S(A))X^{(k)} + X^{(k)}(\alpha_2 I - S(B)) + C, \\
(\beta_1 I + S(A))X^{(k+1)} + X^{(k+1)}(\beta_2 I + S(B)) \\
\qquad = (\beta_1 I - P(A))X^{(k+\frac{1}{2})} + X^{(k+\frac{1}{2})}(\beta_2 I - P(B)) + C,
\end{cases}
\tag{2}
$$

*where $\alpha_j (j = 1, 2)$ are given non-negative constants and $\beta_j (j = 1, 2)$ are positive constants, $I$ is the identity matrix with the appropriate dimension.*

Let $P(A)$, $P(B)$ and $S(A)$, $S(B)$ be the positive definite and skew-Hermitian parts of matrices $A$ and $B$, respectively.

Let

$$
\begin{aligned}
\lambda_{\max}^{(P(A))} &= \max_{\lambda_j \in sp(P(A))} \left\{ |\lambda_j| \right\}, & \mu_{\max}^{(P(B))} &= \max_{\mu_k \in sp(P(B))} \left\{ |\mu_k| \right\}, \\
\lambda_{\min}^{(P(A))} &= \min_{\lambda_j \in sp(P(A))} \left\{ |\lambda_j| \right\}, & \mu_{\min}^{(P(B))} &= \min_{\mu_k \in sp(P(B))} \left\{ |\mu_k| \right\}, \\
\xi_{\max}^{(S(A))} &= \max_{i\xi_j \in sp(S(A))} \left\{ |\xi_j| \right\}, & \zeta_{\max}^{(S(B))} &= \max_{i\zeta_k \in sp(S(B))} \left\{ |\zeta_k| \right\}, \\
\xi_{\min}^{(S(A))} &= \min_{i\xi_j \in sp(S(A))} \left\{ |\xi_j| \right\}, & \zeta_{\min}^{(S(B))} &= \min_{i\zeta_k \in sp(S(B))} \left\{ |\zeta_k| \right\},
\end{aligned}
$$

with $i = \sqrt{-1}$ and

$$
\begin{aligned}
\Theta_{\max} &= \lambda_{\max}^{(P(A))} + \mu_{\max}^{(P(B))}, & Y_{\max} &= \xi_{\max}^{(S(A))} + \zeta_{\max}^{(S(B))}, \\
\Theta_{\min} &= \lambda_{\min}^{(P(A))} + \mu_{\min}^{(P(B))}, & Y_{\min} &= \xi_{\min}^{(S(A))} + \zeta_{\min}^{(S(B))}.
\end{aligned}
$$

In addition, let $\mathbf{A} = \mathbf{P} + \mathbf{S}$, in which

$$
\mathbf{P} = P(\mathbf{A}) = I \otimes P(A) + P(B)^T \otimes I, \quad \mathbf{S} = S(\mathbf{A}) = I \otimes S(A) + S(B)^T \otimes I.
\tag{3}
$$

According to [18], $\Theta_{\max}$, $Y_{\max}$ and $\Theta_{\min}$, $Y_{\min}$ are the upper and lower bounds of the eigenvalues of matrices $\mathbf{P}$ and $\mathbf{S}$, respectively.

The convergence theorem of the FPPSS iterative method for solving the continuous Sylvester Equation (1) is proved as follows.

**Theorem 2.** *Suppose $A \in \mathbb{C}^{m \times m}$ and $B \in \mathbb{C}^{n \times n}$ are positive semidefinite matrices, and at least one of them is a positive definite matrix. $\alpha_j (j = 1, 2)$ are given non-negative constants and $\beta_j (j = 1, 2)$ are positive constants. Let:*

$$
M(\alpha, \beta) = (\beta I + \mathbf{S})^{-1}(\beta I - \mathbf{P})(\alpha I + \mathbf{P})^{-1}(\alpha I - \mathbf{S}),
\tag{4}
$$

*and*

$$
\alpha = \alpha_1 + \alpha_2, \ \beta = \beta_1 + \beta_2,
\tag{5}
$$

*then the upper bound of the spectral radius $\rho(M(\alpha, \beta))$ of the iterative matrix (4) of the iterative method (2) is:*

$$
\sigma(\alpha, \beta) = \max_{\Theta} \left| \frac{\beta - \Theta}{\alpha + \Theta} \right| \cdot \max_Y \sqrt{\frac{\alpha^2 + Y^2}{\beta^2 + Y^2}}.
\tag{6}
$$

*In the meantime, if parameters $\alpha$ and $\beta$ satisfy:*

$$(\alpha, \beta) \in \bigcup_{\ell=1}^{4} \Omega_\ell, \tag{7}$$

*with*

$$
\begin{aligned}
\Omega_1 &= \{ (\alpha, \beta) | \alpha \leq \beta \leq \beta^*(\alpha) \}, \\
\Omega_2 &= \{ (\alpha, \beta) | \beta \geq \max\{\alpha, \beta^*(\alpha)\}, \phi_1(\alpha, \beta) > 0 \}, \\
\Omega_3 &= \{ (\alpha, \beta) | \beta^*(\alpha) \leq \beta \leq \alpha \}, \\
\Omega_4 &= \{ (\alpha, \beta) | \beta < \min\{\alpha, \beta^*(\alpha)\}, \phi_2(\alpha, \beta) > 0 \},
\end{aligned}
$$

*where functions $\phi_1(\alpha, \beta)$, $\phi_2(\alpha, \beta)$ and $\beta^*(\alpha)$ are as follows:*

$$
\begin{aligned}
\phi_1(\alpha, \beta) &= (\beta - \alpha)\left(\Theta_{\min}^2 - Y_{\max}^2\right) + 2\alpha\beta\Theta_{\min} + 2Y_{\max}^2\Theta_{\min}, \\
\phi_2(\alpha, \beta) &= (\beta - \alpha)\left(\Theta_{\max}^2 - Y_{\min}^2\right) + 2\alpha\beta\Theta_{\max} + 2Y_{\min}^2\Theta_{\max}, \\
\beta^*(\alpha) &= \frac{\alpha(\Theta_{\max} + \Theta_{\min}) + 2\Theta_{\max}\Theta_{\min}}{2\alpha + \Theta_{\max} + \Theta_{\min}} \in [\Theta_{\min}, \Theta_{\max}],
\end{aligned}
\tag{8}
$$

*we can prove that $\sigma(\alpha, \beta) < 1$, that is, the FPPSS iterative method (2) converges to the exact solution $X^*$ of the continuous Sylvester Equation (1).*

**Proof.** By Kronecker product, the FPPSS iterative method (2) can be transformed into

$$
\begin{cases}
\left[I \otimes (\alpha_1 I + P(A)) + (\alpha_2 I + P(B))^T \otimes I\right] x^{(k+\frac{1}{2})} \\
\qquad = \left[I \otimes (\alpha_1 I - S(A)) + (\alpha_2 I - S(B))^T \otimes I\right] x^{(k)} + c, \\
\left[I \otimes (\beta_1 I + S(A)) + (\beta_2 I + S(B))^T \otimes I\right] x^{(k+1)} \\
\qquad = \left[I \otimes (\beta_1 I - P(A)) + (\beta_2 I - P(B))^T \otimes I\right] x^{(k+\frac{1}{2})} + c,
\end{cases}
\tag{9}
$$

and Equation (9) can be further turned into:

$$
\begin{cases}
\left[(\alpha_1 + \alpha_2) I + I \otimes P(A) + P(B)^T \otimes I\right] x^{(k+\frac{1}{2})} \\
\qquad = \left[(\alpha_1 + \alpha_2) I - I \otimes S(A) - S(B)^T \otimes I\right] x^{(k)} + c, \\
\left[(\beta_1 + \beta_2) I + I \otimes S(A) + S(B)^T \otimes I\right] x^{(k+1)} \\
\qquad = \left[(\beta_1 + \beta_2) I - I \otimes P(A) - P(B)^T \otimes I\right] x^{(k+\frac{1}{2})} + c,
\end{cases}
\tag{10}
$$

which can be rewritten equivalently as:

$$
\begin{cases}
[\alpha I + \mathbf{P}] x^{(k+\frac{1}{2})} = [\alpha I - \mathbf{S}] x^{(k)} + c, \\
[\beta I + \mathbf{S}] x^{(k+1)} = [\beta I - \mathbf{P}] x^{(k+\frac{1}{2})} + c.
\end{cases}
\tag{11}
$$

After the Formula (11) is reorganized, we can get:

$$
\begin{aligned}
x^{(k+1)} &= \left[(\beta I + \mathbf{S})^{-1}(\beta I - \mathbf{P})(\alpha I + \mathbf{P})^{-1}(\alpha I - \mathbf{S})\right] x^{(k)} \\
&\quad + \left[(\alpha + \beta)(\beta I + \mathbf{S})^{-1}(\alpha I + \mathbf{P})^{-1}\right] c \\
&= M(\alpha, \beta) x^{(k)} + N(\alpha, \beta) c,
\end{aligned}
\tag{12}
$$

where $M(\alpha, \beta)$ is an iterative matrix.

According to the [19], $\mathbf{P}$ is a positive definite matrix, $\mathbf{S}$ is a Skew-Hermitian matrix, $\alpha$ is a non-negative constant, and $\beta$ is a normal number.

The spectral radius of the iterative matrix $M(\alpha, \beta)$ satisfies:

$$\begin{aligned} \rho(M(\alpha,\beta)) \quad &= \rho\Big((\beta I + \mathbf{S})^{-1}(\beta I - \mathbf{P})(\alpha I + \mathbf{P})^{-1}(\alpha I - \mathbf{S})\Big) \\ &\leq \|(\beta I + \mathbf{S})^{-1}(\beta I - \mathbf{P})(\alpha I + \mathbf{P})^{-1}(\alpha I - \mathbf{S})\|_2. \end{aligned} \tag{13}$$

Because

$$(\beta I + \mathbf{S})^{-1}(\beta I - \mathbf{P})(\alpha I + \mathbf{P})^{-1}(\alpha I - \mathbf{S})$$

is similar to:

$$(\beta I - \mathbf{P})(\alpha I + \mathbf{P})^{-1}(\alpha I - \mathbf{S})(\beta I + \mathbf{S})^{-1},$$

(13) can be rewritten as:

$$\begin{aligned} \rho(M(\alpha,\beta)) \quad &= \rho\Big((\beta I + \mathbf{S})^{-1}(\beta I - \mathbf{P})(\alpha I + \mathbf{P})^{-1}(\alpha I - \mathbf{S})\Big) \\ &\leq \|(\beta I - \mathbf{P})(\alpha I + \mathbf{P})^{-1}(\alpha I - \mathbf{S})(\beta I + \mathbf{S})^{-1}\|_2 \\ &\leq \|(\beta I - \mathbf{P})(\alpha I + \mathbf{P})^{-1}\|_2 \|(\alpha I - \mathbf{S})(\beta I + \mathbf{S})^{-1}\|_2 \\ &= \|V_1(\alpha)\|_2 \|V_2(\alpha)\|_2. \end{aligned} \tag{14}$$

(1) Consider: $\|V_1(\alpha)\|_2 = \|(\beta I - \mathbf{P})(\alpha I + \mathbf{P})^{-1}\|_2$

$$\begin{aligned} \|V_1(\alpha)\|_2^2 \quad &= \max\lambda\left\{ \left[(\beta I - \mathbf{P})(\alpha I + \mathbf{P})^{-1}\right]^T \left[(\beta I - \mathbf{P})(\alpha I + \mathbf{P})^{-1}\right] \right\} \\ &= \max\lambda\left\{ (\alpha I + \mathbf{P})^{-T}(\beta I - \mathbf{P})^T(\beta I - \mathbf{P})(\alpha I + \mathbf{P})^{-1} \right\} \\ &= \max\lambda\left\{ \left[(\alpha I + \mathbf{P})^T\right]^{-1}(\beta I - \mathbf{P})^T(\beta I - \mathbf{P})(\alpha I + \mathbf{P})^{-1} \right\}, \end{aligned} \tag{15}$$

for

$$\left[(\alpha I + \mathbf{P})^T\right]^{-1}(\beta I - \mathbf{P})^T(\beta I - \mathbf{P})(\alpha I + \mathbf{P})^{-1}$$

is similar to:

$$(\beta I - \mathbf{P})^T(\beta I - \mathbf{P})(\alpha I + \mathbf{P})^{-1}\left[(\alpha I + \mathbf{P})^T\right]^{-1},$$

(15) can be rewritten as:

$$\begin{aligned} \|V_1(\alpha)\|_2^2 \quad &= \max\lambda\left\{ (\beta I - \mathbf{P})^T(\beta I - \mathbf{P})(\alpha I + \mathbf{P})^{-1}\left[(\alpha I + \mathbf{P})^T\right]^{-1} \right\} \\ &= \max\lambda\left\{ \left[(\beta I - \mathbf{P})^T(\beta I - \mathbf{P})\right]\left[(\alpha I + \mathbf{P})^T(\alpha I + \mathbf{P})\right]^{-1} \right\} \\ &= \max\lambda\left\{ \left[(\beta I - \mathbf{P}^T)(\beta I - \mathbf{P})\right]\left[(\alpha I + \mathbf{P}^T)(\alpha I + \mathbf{P})\right]^{-1} \right\} \\ &= \max\lambda\left\{ \left[\beta^2 I - \beta(\mathbf{P} + \mathbf{P}^T) + \mathbf{P}^T\mathbf{P}\right]\left[\alpha^2 I + \alpha(\mathbf{P} + \mathbf{P}^T) + \mathbf{P}^T\mathbf{P}\right]^{-1} \right\}. \end{aligned} \tag{16}$$

(16) can be equivalently rewritten as

$$\begin{aligned} \|V_1(\alpha)\|_2^2 \quad &= \max\frac{\lambda\left[\beta^2 I - \beta(\mathbf{P}+\mathbf{P}^T) + \mathbf{P}^T\mathbf{P}\right]}{\lambda\left[\alpha^2 I + \alpha(\mathbf{P}+\mathbf{P}^T) + \mathbf{P}^T\mathbf{P}\right]} \\ &= \max\frac{\beta^2 - 2\beta\Theta + \Theta^2}{\alpha^2 + 2\alpha\Theta + \Theta^2} \\ &= \max\left(\frac{\beta - \Theta}{\alpha + \Theta}\right)^2, \end{aligned} \tag{17}$$

so $\|V_1(\alpha)\|_2 = \max\limits_{\Theta}\left|\frac{\beta - \Theta}{\alpha + \Theta}\right|$.

(2) Consider $\|V_2(\alpha)\|_2 = \|(\alpha I - \mathbf{S})(\beta I + \mathbf{S})^{-1}\|_2$

In the same way as the proof process of $\|V_1(\alpha)\|_2 = \|(\beta I - \mathbf{P})(\alpha I + \mathbf{P})^{-1}\|_2$, we can get

$$
\begin{aligned}
\|V_2(\alpha)\|_2^2 &= \max \frac{\lambda\left[\alpha^2 I - \alpha\left(\mathbf{S} + \mathbf{S}^T\right) + \mathbf{S}^T \mathbf{S}\right]}{\lambda\left[\beta^2 I + \beta\left(\mathbf{S} + \mathbf{S}^T\right) + \mathbf{S}^T \mathbf{S}\right]} \\
&= \max \frac{\alpha^2 + Y^2}{\beta^2 + Y^2},
\end{aligned}
\tag{18}
$$

so $\|V_2(\alpha)\| = \max\limits_{Y} \sqrt{\frac{\alpha^2 + Y^2}{\beta^2 + Y^2}}$.

Bring (17) and (18) into (14), we get:

$$
\rho(M(\alpha, \beta)) \leq \max_{\Theta}\left|\frac{\beta - \Theta}{\alpha + \Theta}\right| \cdot \max_{Y} \sqrt{\frac{\alpha^2 + Y^2}{\beta^2 + Y^2}},
\tag{19}
$$

which gives the upper bound $\sigma(\alpha, \beta) = \max\limits_{\Theta}\left|\frac{\beta - \Theta}{\alpha + \Theta}\right| \cdot \max\limits_{Y} \sqrt{\frac{\alpha^2 + Y^2}{\beta^2 + Y^2}}$ of the spectral radius of the iterative matrix $M(\alpha, \beta)$.

In the following, similar to the Theorem 2.2 of the literature [20] to prove the process idea, we can get:

$$
\max_{\Theta}\left|\frac{\beta - \Theta}{\alpha + \Theta}\right| = \max\left\{\left|\frac{\beta - \Theta_{\max}}{\alpha + \Theta_{\max}}\right|, \left|\frac{\beta - \Theta_{\min}}{\alpha + \Theta_{\min}}\right|\right\},
\tag{20}
$$

absorb the absolute value symbol on the right side of (20) to get:

$$
\frac{\beta - \Theta_{\min}}{\alpha + \Theta_{\min}} = \frac{\Theta_{\max} - \beta}{\alpha + \Theta_{\max}}.
\tag{21}
$$

It can be solved from the Formula (21) that:

$$
\beta^*(\alpha) = \frac{\alpha(\Theta_{\max} + \Theta_{\min}) + 2\Theta_{\max}\Theta_{\min}}{2\alpha + \Theta_{\max} + \Theta_{\min}} \in [\Theta_{\min}, \Theta_{\max}].
\tag{22}
$$

Simultaneously we have:

$$
\begin{aligned}
\|V_1(\alpha)\|_2 &= \max_{\Theta}\left|\frac{\beta - \Theta}{\alpha + \Theta}\right| \\
&= \begin{cases} \frac{\Theta_{\max} - \beta}{\Theta_{\max} + \alpha}, & \beta < \beta^*(\alpha), \\ \frac{\beta - \Theta_{\min}}{\Theta_{\min} + \alpha}, & \beta \geq \beta^*(\alpha). \end{cases}
\end{aligned}
\tag{23}
$$

The same reason can be used to obtained:

$$
\begin{aligned}
\|V_2(\alpha)\| &= \max_{Y} \sqrt{\frac{\alpha^2 + Y^2}{\beta^2 + Y^2}} \\
&= \begin{cases} \sqrt{\frac{\alpha^2 + Y_{\max}^2}{\beta^2 + Y_{\max}^2}}, & \alpha \leq \beta, \\ \sqrt{\frac{\alpha^2 + Y_{\min}^2}{\beta^2 + Y_{\min}^2}}, & \alpha > \beta. \end{cases}
\end{aligned}
\tag{24}
$$

At this point we can divide the area $\Omega = \{(\alpha, \beta) | \alpha \geq 0, \beta > 0\}$ into the following four parts according to (23) and (24):

$$
\begin{aligned}
\Omega_1 &= \{(\alpha, \beta) | \alpha \leq \beta < \beta^*(\alpha)\}, \quad \Omega_2 = \{(\alpha, \beta) | \beta \geq \max\{\alpha, \beta^*(\alpha)\}\}, \\
\Omega_3 &= \{(\alpha, \beta) | \beta^*(\alpha) \leq \beta \leq \alpha\}, \quad \Omega_4 = \{(\alpha, \beta) | \beta < \min\{\alpha, \beta^*(\alpha)\}\}.
\end{aligned}
$$

From (19), (23), and (24) we can know:

(1) For $(\alpha, \beta) \in \Omega_1 = \{ (\alpha, \beta) | \alpha \leq \beta \leq \beta^*(\alpha) \}$,

$$\rho(M(\alpha, \beta)) \leq \frac{\Theta_{\max} - \beta}{\Theta_{\max} + \alpha} \cdot \sqrt{\frac{\alpha^2 + Y_{\max}^2}{\beta^2 + Y_{\max}^2}} < 1.$$

(2) For $(\alpha, \beta) \in \Omega_2 = \{ (\alpha, \beta) | \beta \geq \max\{\alpha, \beta^*(\alpha)\} \}$,

$$\rho(M(\alpha, \beta)) \leq \frac{\beta - \Theta_{\min}}{\Theta_{\min} + \alpha} \cdot \sqrt{\frac{\alpha^2 + Y_{\max}^2}{\beta^2 + Y_{\max}^2}}, \tag{25}$$

to make (25) less than 1, if and only if

$$\phi_1(\alpha, \beta) = (\beta - \alpha)\left(\Theta_{\min}^2 - Y_{\max}^2\right) + 2\alpha\beta\Theta_{\min} + 2Y_{\max}^2\Theta_{\min} > 0.$$

(3) For $(\alpha, \beta) \in \Omega_3 = \{ (\alpha, \beta) | \beta^*(\alpha) \leq \beta \leq \alpha \}$,

$$\rho(M(\alpha, \beta)) \leq \frac{\beta - \Theta_{\min}}{\alpha + \Theta_{\min}} \cdot \sqrt{\frac{\alpha^2 + Y_{\min}^2}{\beta^2 + Y_{\min}^2}} < \frac{\beta}{\alpha} \cdot \frac{\alpha}{\beta} = 1.$$

(4) For $(\alpha, \beta) \in \Omega_4 = \{ (\alpha, \beta) | \beta < \min\{\alpha, \beta^*(\alpha)\} \}$,

$$\rho(M(\alpha, \beta)) \leq \frac{\Theta_{\max} - \beta}{\Theta_{\max} + \alpha} \cdot \sqrt{\frac{\alpha^2 + Y_{\min}^2}{\beta^2 + Y_{\min}^2}}, \tag{26}$$

to make (26) less than 1, if and only if:

$$\phi_2(\alpha, \beta) = (\beta - \alpha)\left(\Theta_{\max}^2 - Y_{\min}^2\right) + 2\alpha\beta\Theta_{\max} + 2Y_{\min}^2\Theta_{\max} > 0.$$

In summary, we can draw the conclusion:

$$\rho(M(\alpha, \beta)) \leq \sigma(\alpha, \beta) < 1, \ \forall (\alpha, \beta) \in \bigcup_{\ell=1}^{4} \Omega_\ell.$$

Theorem 2 is verified. $\square$

**Theorem 3.** *The theoretical optimal parameter that makes $\sigma(\alpha, \beta)$ the minimum is:*

$$(\alpha^*, \beta^*) = \operatorname*{argmin}_{\alpha, \beta}\{\sigma(\alpha, \beta)\} = \begin{cases} (\alpha_1, \beta^*(\alpha_1)), & \Theta_{\max}\Theta_{\min} \leq Y_{\min}^2, \\ (\alpha_0, \beta^*(\alpha_0)), & Y_{\min}^2 < \Theta_{\max}\Theta_{\min} < Y_{\max}^2, \\ (\alpha_2, \beta^*(\alpha_2)), & \Theta_{\max}\Theta_{\min} \geq Y_{\max}^2, \end{cases}$$

*where*

$$\alpha_1 = \frac{Y_{\min}^2 - \Theta_{\max}\Theta_{\min} + \sqrt{\left(Y_{\min}^2 + \Theta_{\max}^2\right)\left(Y_{\min}^2 + \Theta_{\min}^2\right)}}{\Theta_{\max} + \Theta_{\min}},$$
$$\alpha_0 = \sqrt{\Theta_{\max}\Theta_{\min}},$$
$$\alpha_2 = \frac{Y_{\max}^2 - \Theta_{\max}\Theta_{\min} + \sqrt{\left(Y_{\max}^2 + \Theta_{\max}^2\right)\left(Y_{\max}^2 + \Theta_{\min}^2\right)}}{\Theta_{\max} + \Theta_{\min}}.$$

*The upper bound of the spectral radius of the corresponding iterative matrix is:*

$$\sigma(\alpha^*, \beta^*) = \begin{cases} \sigma(\alpha_1), & \Theta_{\max}\Theta_{\min} \leq Y_{\min}^2, \\ \sigma(\alpha_0), & Y_{\min}^2 < \Theta_{\max}\Theta_{\min} < Y_{\max}^2, \\ \sigma(\alpha_2), & \Theta_{\max}\Theta_{\min} \geq Y_{\max}^2, \end{cases}$$

*where*

$$\sigma(\alpha) = \sigma(\alpha, \beta^*(\alpha)) = \begin{cases} \frac{\beta^*(\alpha) - \Theta_{\min}}{\alpha + \Theta_{\min}} \cdot \sqrt{\frac{\alpha^2 + Y_{\min}^2}{\beta^*(\alpha)^2 + Y_{\min}^2}}, & \alpha > \alpha_0, \\[2ex] \frac{\beta^*(\alpha) - \Theta_{\min}}{\alpha + \Theta_{\min}} \cdot \sqrt{\frac{\alpha^2 + Y_{\max}^2}{\beta^*(\alpha)^2 + Y_{\max}^2}}, & \alpha \leq \alpha_0. \end{cases}$$

**Proof.** From (23) and (24) we can know:

$$\sigma(\alpha, \beta) = \begin{cases} \frac{\Theta_{\max} - \beta}{\Theta_{\max} + \alpha} \cdot \sqrt{\frac{\alpha^2 + Y_{\max}^2}{\beta^2 + Y_{\max}^2}}, & (\alpha, \beta) \in \Omega_1, \\[2ex] \frac{\beta - \Theta_{\min}}{\Theta_{\min} + \alpha} \cdot \sqrt{\frac{\alpha^2 + Y_{\max}^2}{\beta^2 + Y_{\max}^2}}, & (\alpha, \beta) \in \Omega_2, \\[2ex] \frac{\beta - \Theta_{\min}}{\alpha + \Theta_{\min}} \cdot \sqrt{\frac{\alpha^2 + Y_{\min}^2}{\beta^2 + Y_{\min}^2}}, & (\alpha, \beta) \in \Omega_3, \\[2ex] \frac{\Theta_{\max} - \beta}{\Theta_{\max} + \alpha} \cdot \sqrt{\frac{\alpha^2 + Y_{\min}^2}{\beta^2 + Y_{\min}^2}}, & (\alpha, \beta) \in \Omega_4, \end{cases} \tag{27}$$

we can observe from (27) that $\sigma'_\beta(\alpha, \beta) < 0$ when $(\alpha, \beta) \in \Omega_1$ and $(\alpha, \beta) \in \Omega_4$, $\sigma'_\beta(\alpha, \beta) > 0$ when $(\alpha, \beta) \in \Omega_2$ and $(\alpha, \beta) \in \Omega_3$, then at $\beta = \beta^*(\alpha)$, $\sigma'_\beta(\alpha, \beta)$ has a minimum value and is also the minimum value.

Substitute $\beta = \beta^*(\alpha)$ into (27), then:

$$\sigma(\alpha) = \sigma(\alpha, \beta^*(\alpha)) = \begin{cases} \frac{\beta^*(\alpha) - \Theta_{\min}}{\alpha + \Theta_{\min}} \cdot \sqrt{\frac{\alpha^2 + Y_{\min}^2}{\beta^*(\alpha)^2 + Y_{\min}^2}}, & \alpha > \alpha_0, \\[2ex] \frac{\beta^*(\alpha) - \Theta_{\min}}{\alpha + \Theta_{\min}} \cdot \sqrt{\frac{\alpha^2 + Y_{\max}^2}{\beta^*(\alpha)^2 + Y_{\max}^2}}, & \alpha \leq \alpha_0, \end{cases} \tag{28}$$

obviously, computing the minimum value of (27) is converted to solving the minimum value of (28).

Find the derivative number for (28) and get:

$$\sigma'(\alpha) = \begin{cases} c_1(\alpha)\eta_1(\alpha), & \alpha > \alpha_0, \\ c_2(\alpha)\eta_2(\alpha), & \alpha < \alpha_0, \end{cases} \tag{29}$$

where $c_1(\alpha)$ and $c_2(\alpha)$ are two positive function, and:

$$\begin{aligned} \eta_1(\alpha) &= (\Theta_{\max} + \Theta_{\min})\alpha^2 + 2\alpha(\Theta_{\max}\Theta_{\min} - Y_{\min}^2) - Y_{\min}^2(\Theta_{\max} + \Theta_{\min}), \\ \eta_2(\alpha) &= (\Theta_{\max} + \Theta_{\min})\alpha^2 + 2\alpha(\Theta_{\max}\Theta_{\min} - Y_{\max}^2) - Y_{\max}^2(\Theta_{\max} + \Theta_{\min}). \end{aligned} \tag{30}$$

It can be observed that $\eta_1(\alpha)$ is similar to $\eta_2(\alpha)$ format and has a positive root and a negative root. The positive roots are denoted as $\alpha_1$ and $\alpha_2$, respectively, and because of $Y_{\max} > Y_{\min}$, $\alpha_1 < \alpha_2$. Also note that $\Theta_{\max} + \Theta_{\min} \geq 0$.

Bring $\alpha = \alpha_0$ into (30) to get:

$$\begin{aligned} \eta_1(\alpha_0) &= (\sqrt{\Theta_{\max}} + \sqrt{\Theta_{\min}})^2(\Theta_{\max}\Theta_{\min} - Y_{\min}^2), \\ \eta_2(\alpha_0) &= (\sqrt{\Theta_{\max}} + \sqrt{\Theta_{\min}})^2(\Theta_{\max}\Theta_{\min} - Y_{\max}^2). \end{aligned} \tag{31}$$

According to (31), we can find:

(1) When $\Theta_{\max}\Theta_{\min} \leq Y_{\min}^2$, we have $\eta_1(\alpha_0) < 0$ and $\eta_2(\alpha_0) < 0$, then there are $\alpha_0 < \alpha_1 < \alpha_2$, at this time $\sigma(\alpha, \beta)$ takes the minimum at $(\alpha_1, \beta^*(\alpha_1))$.

(2) When $Y_{\min}^2 < \Theta_{\max}\Theta_{\min} < Y_{\max}^2$, we have $\eta_1(\alpha_0) > 0$ and $\eta_2(\alpha_0) < 0$, then there are $\alpha_1 < \alpha_0 < \alpha_2$, at this time $\sigma(\alpha, \beta)$ takes the minimum at $(\alpha_0, \beta^*(\alpha_0))$.

(3) When $\Theta_{\max}\Theta_{\min} \geq Y_{\max}^2$, we have $\eta_1(\alpha_0) > 0$ and $\eta_2(\alpha_0) > 0$, then there are $\alpha_1 < \alpha_2 < \alpha_0$, at this time $\sigma(\alpha, \beta)$ takes the minimum at $(\alpha_2, \beta^*(\alpha_2))$.

In summary, Theorem 3 is verified. $\square$

## 3. Numerical Experiments

In this part, we use numerical experiments to compare the FPPSS iterative method, PSS iterative method and HSS iterative method for solving the continuous Sylvester Equation (1) in term of iteration steps (IT) and computing time (CPU).

In the implementation of the algorithm, for the convenience of calculation, the initial matrix $X^{(0)}$ is taken as a zero matrix, and the iterative stopping criterion is $\frac{\|C - AX^{(k)} - X^{(k)}B\|_F}{\|C\|_F} \leq 10^{-6}$. In addition, in each step of the iterative method, the subproblem is solved by the direct method in [20].

**Example 1.** *In order to generate large and sparse matrices A and B, we established them in the following ways which can also be seen in [13]:*

$$
A = \begin{pmatrix}
10 & 1 & & & 1 \\
2 & 10 & 1 & & \\
& \ddots & \ddots & \ddots & \\
& & 2 & 10 & 1 \\
1 & & & 2 & 10
\end{pmatrix}, B = \begin{pmatrix}
8 & 1 & & & 1 \\
3 & 8 & 1 & & \\
& \ddots & \ddots & \ddots & \\
& & 3 & 8 & 1 \\
1 & & & 3 & 8
\end{pmatrix}.
$$

Tables 1 and 2 lists the numerical results of FPPSS, PSS, and HSS iterative method using experimental optimal iterative parameters. $\alpha_1^*$, $\beta_1^*$, and $\alpha^*$ (where $\beta^* = \alpha^*$) represent the experimental quasi-optimal parameters of the FPPSS, PSS, and HSS iterative methods, respectively.

**Table 1.** IT and CPU for four parameters positive and skew-Hermitian splitting (FPPSS), positive and skew-Hermitian splitting (PSS), and Hermitian and skew-Hermitian splitting iterative method (HSS) for Example 1 when using experimental quasi-optimal parameters.

| Method | FPPSS | | PSS | | HSS | |
|---|---|---|---|---|---|---|
| $n$ | IT | CPU | IT | CPU | IT | CPU |
| $n = 8$ | 6 | 1.312 | 16 | 1.153 | 15 | 1.249 |
| $n = 16$ | 6 | 1.318 | 16 | 1.147 | 16 | 1.166 |
| $n = 32$ | 6 | 1.332 | 17 | 1.298 | 16 | 1.250 |
| $n = 64$ | 6 | 1.424 | 17 | 1.559 | 16 | 1.495 |
| $n = 128$ | 6 | 2.230 | 17 | 3.134 | 16 | 3.655 |
| $n = 256$ | 6 | 8.406 | 17 | 19.187 | 16 | 26.811 |
| $n = 512$ | 6 | 86.889 | 17 | 192.139 | 16 | 230.645 |
| $n = 1024$ | 6 | 818.956 | - | - | - | - |

**Table 2.** The practical optimal value for FPPSS, PSS, and HSS for Example 1.

| Method | FPPSS | | PSS | HSS |
|---|---|---|---|---|
| $n$ | $\alpha_1^*$ | $\beta_1^*$ | $\alpha^* = \beta^*$ | $\alpha^* = \beta^*$ |
| $n = 8$ | 0.3794 | 9 | 4.7843 | 4.7843 |
| $n = 16$ | 0.4275 | 9 | 4.7750 | 4.7750 |
| $n = 32$ | 0.4402 | 9 | 4.7714 | 4.7714 |
| $n = 64$ | 0.4434 | 9 | 4.7702 | 4.7702 |
| $n = 128$ | 0.4442 | 9 | 4.7698 | 4.7698 |
| $n = 256$ | 0.4444 | 9 | 4.7697 | 4.7697 |
| $n = 512$ | 0.4444 | 9 | 4.7697 | 4.7697 |
| $n = 1024$ | 0.4444 | 9 | - | - |

**Example 2.** *The continuous Sylvester equation (1) with $m = n$ and the matrices:*

$$\begin{cases} A = diag(1, 2, \ldots, n) + 10^{-3}L^{\mathrm{T}}, \\ B = 2^{-t}I + diag(1, 2, \ldots n) + 10^{-3}L^{\mathrm{T}} + 2^{-t}L, \end{cases}$$

*with L the strictly lower triangular matrix having ones in the lower triangle part and t is a problem parameter to be specified in actual computations.*

Tables 3 and 4 lists the numerical results of FPPSS, PSS, and HSS iterative method using experimental optimal iterative parameters. $\alpha_1^*$, $\beta_1^*$, and $\alpha^*$ (where $\beta^* = \alpha^*$) represent the experimental quasi-optimal parameters of the FPPSS, PSS, and HSS iterative methods, respectively.

**Table 3.** IT and CPU for FPPSS, PSS, and HSS for Example 2 when using experimental quasi-optimal parameters.

| Method | FPPSS | | PSS | | HSS | |
|---|---|---|---|---|---|---|
| $n$ | IT | CPU | IT | CPU | IT | CPU |
| $n = 8$ | 2 | 1.280 | 40 | 1.132 | 30 | 1.368 |
| $n = 16$ | 3 | 1.335 | 54 | 1.317 | 44 | 1.200 |
| $n = 32$ | 3 | 1.399 | 73 | 1.702 | 65 | 1.443 |
| $n = 64$ | 3 | 1.383 | 100 | 3.305 | 93 | 3.493 |
| $n = 128$ | 4 | 1.863 | 139 | 22.177 | 134 | 27.289 |
| $n = 256$ | 5 | 6.862 | 196 | 361.342 | 191 | 438.372 |
| $n = 512$ | 6 | 69.461 | - | - | - | - |
| $n = 1024$ | 8 | 1122.085 | - | - | - | - |

**Table 4.** The practical optimal value for FPPSS, PSS, and HSS for Example 2.

| Method | FPPSS | | PSS | HSS |
|---|---|---|---|---|
| $n$ | $\alpha_1^*$ | $\beta_1^*$ | $\alpha^* = \beta^*$ | $\alpha^* = \beta^*$ |
| $n = 8$ | $9.9114 \times 10^{-6}$ | 2.5500 | 1.4142 | 1.4142 |
| $n = 16$ | $3.7485 \times 10^{-5}$ | 2.7500 | 2.0000 | 2.0000 |
| $n = 32$ | $1.4448 \times 10^{-4}$ | 2.8678 | 2.8284 | 2.8284 |
| $n = 64$ | $5.6589 \times 10^{-4}$ | 2.9323 | 4.0000 | 4.0000 |
| $n = 128$ | 0.0022 | 2.9675 | 5.6569 | 5.6569 |
| $n = 256$ | 0.0089 | 2.9912 | 8.0000 | 8.0000 |
| $n = 512$ | 0.0351 | 3.0259 | - | - |
| $n = 1024$ | 0.1358 | 3.1305 | - | - |

**Example 3.** *Consider Equation (1), where $m = n$, $A = B = M + qN + \frac{100}{(n+1)^2}I$ and $M$, $N \in \mathbb{R}^{n \times n}$ are the following three diagonal matrices.*

$M = \text{tridiag}(-1, -2, -1)$ and $N = \text{tridiag}(0.5, 0, -0.5)$, Tables 5 and 6 list the numerical results of FPPSS and PSS iterative method using experimental optimal iterative parameters. $\alpha_1^*$ (where $\alpha_1^* = \alpha_2^*$), $\beta_1^*$ (where $\beta_1^* = \beta_2^*$) and $\alpha^*$ (where $\beta^* = \alpha^*$) represent the experimental quasi-optimal parameters of the FPPSS, PSS, and HSS iterative methods, respectively.

**Table 5.** IT and CPU for FPPSS, PSS, and HSS for Example 3 when using experimental quasi-optimal parameters.

| Method | | FPPSS | | PSS | | HSS | |
|---|---|---|---|---|---|---|---|
| $q$ | $n$ | IT | CPU | IT | CPU | IT | CPU |
| $q = 1$ | $n = 8$ | 8 | 1.302 | 20 | 1.192 | 28 | 1.124 |
| | $n = 16$ | 15 | 1.344 | 39 | 1.298 | 47 | 1.334 |
| | $n = 32$ | 34 | 1.439 | 81 | 1.673 | 93 | 1.874 |
| | $n = 64$ | 62 | 3.029 | 164 | 6.281 | 203 | 7.193 |
| | $n = 128$ | 104 | 13.175 | - | - | - | - |
| | $n = 256$ | 175 | 256.191 | - | - | - | - |
| $q = 10$ | $n = 8$ | 9 | 1.385 | 23 | 1.268 | 33 | 1.227 |
| | $n = 16$ | 14 | 1.464 | 43 | 1.259 | 60 | 1.436 |
| | $n = 32$ | 21 | 1.501 | 83 | 1.647 | 123 | 2.041 |
| | $n = 64$ | 30 | 2.119 | 166 | 5.409 | 251 | 10.950 |
| | $n = 128$ | 44 | 9.310 | - | - | - | - |
| | $n = 256$ | 67 | 152.779 | - | - | - | - |
| $q = 100$ | $n = 8$ | 11 | 1.544 | 24 | 1.290 | 39 | 1.151 |
| | $n = 16$ | 25 | 1.502 | 43 | 1.472 | 69 | 1.291 |
| | $n = 32$ | 61 | 1.824 | 84 | 1.585 | 133 | 1.900 |
| | $n = 64$ | 91 | 4.032 | 167 | 6.187 | 265 | 10.966 |
| | $n = 128$ | 123 | 25.951 | - | - | - | - |
| | $n = 256$ | 182 | 365.007 | - | - | - | - |

**Table 6.** The practical optimal value for FPPSS, PSS, and HSS for Example 3.

| Method | | FPPSS | | PSS HSS | |
|---|---|---|---|---|---|
| $q$ | $n$ | $\alpha_1^*$ | $\beta_1^*$ | $\alpha^* = \beta^*$ | $\alpha^* = \beta^*$ |
| $q = 1$ | $n = 8$ | 0.2730 | 3.2346 | 1.3163 | 1.3163 |
| | $n = 16$ | 0.4119 | 2.3460 | 0.6010 | 0.6401 |
| | $n = 32$ | 0.4737 | 2.0918 | 0.3209 | 0.3209 |
| | $n = 64$ | 0.4930 | 2.0237 | 0.1617 | 0.1617 |
| | $n = 128$ | 0.4982 | 2.0060 | - | - |
| | $n = 256$ | 0.4995 | 2.0015 | - | - |
| $q = 10$ | $n = 8$ | 3.2346 | 3.2346 | 1.3163 | 1.3163 |
| | $n = 16$ | 2.3460 | 2.3460 | 0.6401 | 0.6401 |
| | $n = 32$ | 2.0918 | 2.0918 | 0.3209 | 0.3209 |
| | $n = 64$ | 2.0237 | 2.0237 | 0.1617 | 0.1617 |
| | $n = 128$ | 2.0060 | 2.0060 | - | - |
| | $n = 256$ | 2.0015 | 2.0015 | - | - |
| $q = 100$ | $n = 8$ | 3.2346 | 3.2346 | 1.3163 | 1.3163 |
| | $n = 16$ | 2.3460 | 2.3460 | 0.6401 | 0.6401 |
| | $n = 32$ | 2.0918 | 2.0918 | 0.3209 | 0.3209 |
| | $n = 64$ | 2.0237 | 2.0237 | 0.1617 | 0.1617 |
| | $n = 128$ | 2.0060 | 2.0060 | - | - |
| | $n = 256$ | 2.0015 | 2.0015 | - | - |

From the above three examples, we can see that although the runtime of FPPSS iteration method is slightly higher than that of previous iteration method when the dimension of coefficient matrices is small, but lower when the dimension is large. And the iteration steps of FPPSS iteration method are less than that of previous iteration method regardless of the dimension of coefficient matrices. When the matrix dimension is high, the results can still be calculated in a shorter runtime and less iteration steps. Through the above numerical experiments, it is proved that FPPSS iteration method is an effective improved algorithm.

## 4. Conclusions

In this paper, a new four-parameter positive and skew-Hermitian iterative method, namely the FPPSS iterative method, is applied to solve the Sylvester equation of the form $AX + XB = C$, which is a generalization of the classical PSS iterative method [7]. This paper proves that when the parameters satisfy certain conditions, the iterative sequence generated by the FPPSS method converges to the unique solution of the Sylvester equation, and the PSS method is a special case of the FPPSS method. We also give the theoretical optimal sum of the parameters that minimize the upper bound of the spectral radius of the iterative matrix. In addition, it can be seen from the experimental data that the FPPSS iterative method is superior to the PSS and HSS iterative method in most cases in CPU and IT, which indicates that the newly constructed FPPSS iterative method is an effective iterative method for solving the Sylvester equation.

**Author Contributions:** Conceptualization, H.-L.S. and Y.-R.L.; methodology, H.-L.S.; software, Y.-R.L.; validation, H.-L.S., Y.-R.L. and X.-H.S.; formal analysis, H.-L.S.; data curation, Y.-R.L.; writing–original draft preparation, H.-L.S.; writing–review & editing, Y.-R.L.; project administration, X.-H.S.; funding acquisition, H.-L.S.

**Funding:** Project supported by the National Natural Science Foundation of China (No. 11371081) and the Natural Science Foundation of Liaoning Province (No. 20170540323).

**Acknowledgments:** The authors would like to express their great thankfulness to the referees for the comments and constructive suggestions, which are valuable in improving the quality of the original paper.

**Conflicts of Interest:** The authors declare no conflict of interest.

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
