# Peer review of "The Four-Parameter PSS Method for Solving the Sylvester Equation"

_mathematics, doi:10.3390/math7010105_

Round 1

Reviewer 1 Report

The authors should emphasize the scientific contribution of the paper:

by extending Introduction, with a proper analysis of the state of the art;

by listing pros and cons of the approach;

by extending the numerical experiments section, where the computational cost should be properly discussed with a suitable comparative analysis

Author Response

Answer

I have revised the paper in accordance with your valuable comments, and you can see new changes in the review mode.

1.      The background and practical significance of studying this issue have been added to introduction section.

2.      I have summarized the methods for solving equations at different times in introduction section.

3.      I have listed the pros and cons of each existing method in the introduction.

4.      Two new numerical examples have been added to analyze the performance of the methods and the dimension of coefficient matrix is raised to 512×512. By comparing the running time and iteration stepsit shows that the new method greatly improves the old method for high dimensional coefficient matrix equation.

Thanks

Finally, thank you very much for your valuable and detailed comments which helped to improve the presentation of this paper.

Reviewer 2 Report

Reading the Introduction of such a manuscript is a pity: the content is not described properly and the language is poor. The mathematics is not precise since the authors use notions which are not defined, so that by the way the manuscript is not self-contained.

 The results are shallow since they are almost photocopy of existing results, and enlarging the number of parameters is of course a way of making a method faster but it is not effective if we do not know in practice how to compute these parameters. Furthermore the numerics are not convincing, given the small size of the problems (in the introduction the authors start by claiming that matrices are large and sparse) and given the very high number of iterations (which show that the presented methods are not effective.

My personal conclusion is that the current manuscript is not appropriate in any international journal.

Author Response

Answer

I have revised the paper in accordance with your valuable comments, and you can see new changes in the review mode.

1.        The word spelling and grammar errors have been modified.

2.        I have defined the concepts covered in this article and adjusted the content that is inappropriate for the description.

3.        The theoretical optimal values of the parameters have been given in Theorem 3. However, because the theoretical optimal parameters are difficult to realize in calculation, the parameters in the example are the practical optimal parameters.

4.        I have raised the coefficient matrices dimension to 512×512 and added two numerical examples. Through three numerical examples, it is obvious that the new method takes less runtime and iteration steps than the old method, and the higher the dimension of coefficient matrix, the more obvious the conclusion is, which shows that the new method is effective.

Thanks

Finally, thank you very much for your valuable and detailed comments which helped to improve the presentation of this paper.

Reviewer 3 Report

In this paper, a new four-parameter PSS iterative method, namely the FPPSS iterative method, is applied to solve the Sylvester equation of the form AX+XB=C, which is a  generalization of the classical PSS iterative method [5].  

Remarks

So, In rows 35-36 is written HSS from [4]. What is name HSS or PSS ?

An additional literature has to be cites and commented Computers and Math with Appl. Vol. 35 (10),  1998, Pages 35-44

In rows 235-238  we are reading: “In addition, it can be seen from the experimental data that the FPPSS iterative method is superior to the PSS iterative method in most cases in CPU and IT, which indicates that the newly constructed  FPPSS iterative method is an effective iterative method for solving the Sylvester equation.”

So, the results described in Table 1 are not enough to prove that the new FPPSS iterative method is an effective one. More examples and experiments needed. Moreover, it is not clear what are the advantages of the FPPSS iterative method? It is not faster than the PSS iterative method (see CPU time in Table 1)

Author Response

Answer

I have revised the paper in accordance with your valuable comments, and you can see new changes in the review mode.

1.        The abbreviations for HSS and PSS have been changed to its full name.

2.        I have added two new examples in the numerical experiment section and increased the dimension of coefficient matrix to 512×512. By comparing the data of three tables, it is obvious that the higher the dimension of matrix, the more obvious the advantages of FPPSS iteration method in running time and iteration steps, and the lower the cost compared with PSS iteration method. This shows that FPPSS iteration method is more advantageous for large sparse coefficient matrix.

Thanks

Finally, thank you very much for your valuable and detailed comments which helped to improve the presentation of this paper.

Round 2

Reviewer 1 Report

The paper can be accepted in its current form.

Author Response

Thank you very much for your valuable and detailed comments which helped to improve the presentation and quality of this paper.

Reviewer 2 Report

With respect to the first version there is some improvement but still I see substantial problems:

1)      The number of iterations is still not satisfactory (preconditioning?) and the problem sizes too small

2)      The language is not accurate: plea etcse add a space before (PSS) in the abstract, please add the between of and positive at line 22, page 1,

3)      The mathematics is not precise: in the literature the splitting is called HSS (here is PSS); in item (III) the authors write positive definitewithout giving the definition (in many books positive definite implies Hermitian but here this is not the case so a precise definition is useful)

4)      Having many parameters is not good: how to find the parameters in practice?

5)      At page 2, line 53, please replace [9] by [9,a] with [a] as reported below

[a] D. Bertaccini, G.H. Golub, S. Serra-Capizzano, C. Tablino-Possio, Preconditioned HSS methods for the solution of non-Hermitian positive definite linear systems and applications to the discrete convection–diffusion equation, Numer. Math. 99 (2005) 441–484.

 In conclusion a major revision is still needed.

Author Response

(The authors gave the same response as above.)

Reviewer 3 Report

the manuscript has been improved and I propose to publish in this form.

Author Response

(The authors gave the same response as above.)

Round 3

Reviewer 2 Report

There is some improvements but not enough.

The authors have still to work on 4 of the  issues of the previous report:

1)      The number of iterations is still not satisfactory (preconditioning?) and the problem sizes too small

2)      The language is not accurate: please check all the manuscript

3)      The mathematics is not precise: in the literature the splitting is called HSS (here is PSS); in item (III) the authors write positive definitewithout giving the definition (in many books positive definite implies Hermitian but here this is not the case so a precise definition is useful)

4)      Having many parameters is not good: how to find the parameters in practice?